# Long-Term Mastication Changed Salivary Metabolomic Profiles

**DOI:** 10.3390/metabo12070660

**Published:** 2022-07-18

**Authors:** Yoji Saeki, Akane Takenouchi, Etsuyo Otani, Minji Kim, Yumi Aizawa, Yasuko Aita, Atsumi Tomita, Masahiro Sugimoto, Takashi Matsukubo

**Affiliations:** 1Central Laboratory, Lotte Co., Ltd., 3-1-1 Numakage, Minami-ku, Saitama 336-8601, Japan; kim_minji@lotte.co.jp; 2Shinjuku Medical Career College, 5 Samon-cho, Shinju-ku, Tokyo 160-0017, Japan; atoe.akanetsuyo@gmail.com; 3Dental Hygiene Course, Ogaki Women’s University College, 1-109 Nishinokawacho, Gifu 503-8554, Japan; otani@ogaki-tandai.ac.jp; 4School of Health and Social Services, Saitama Prefectural University, 820 Sannomiya, Saitama 343-8540, Japan; 5Institute of Medical Science, Tokyo Medical University, 6-1-1 Shinjuku, Shinjuku-ku, Tokyo 160-8402, Japan; azyumi726@gmail.com (Y.A.); y.sakagu12@gmail.com (Y.A.); tomita@tokyo-med.ac.jp (A.T.); mshrsgmt@gmail.com (M.S.); 6Tokyo Dental College, 2-9-18 Misaki-cho, Chiyoda-ku, Tokyo 101-0061, Japan; matukubo@tdc.ac.jp

**Keywords:** mastication, saliva, metabolomic profile, chewing, liquid chromatography–mass spectrometry

## Abstract

Saliva is an ideal biofluid for monitoring oral and systemic health. Repeated mastication is a typical physical stimulus that improves salivary flow and oral hygiene. Recent metabolomic studies have shown the potential of salivary metabolomic components for various disease monitoring systems. Here, we evaluated the effect of long-term mastication on salivary metabolomic profiles. Young women with good oral hygiene (20.8 ± 0.3 years, *n* = 17) participated. They were prohibited from chewing gum during control periods (4 weeks each) and were instructed to chew a piece of gum base seven times a day for 10 min each time during the intervention period. Paired samples of unstimulated whole saliva collected on the last day of the control and intervention period were compared. Liquid chromatography–time-of-flight mass spectrometry successfully quantified 85 metabolites, of which 41 showed significant differences (*p <* 0.05, Wilcoxon paired test corrected by false discovery rate). Except for a few metabolites, such as citrate, most metabolites showed lower concentrations after the intervention. The pathways related to glycogenic amino acids, such as alanine, arginine, and glutamine, altered considerably. This study suggests that long-term mastication induces unstimulated salivary component-level changes.

## 1. Introduction

Salivary flow and composition are prominent features that reflect oral and systemic health status. Both intrinsic and extrinsic factors affect salivary flow and composition [1]. Therefore, the molecular composition of saliva has been intensively analyzed to explore new molecular biomarkers to monitor these systematic statuses [2,3].

Occlusal masticatory stimulation plays an essential role in patients’ quality of life. Saliva is crucial in chewing, and its volume and chemical properties reflect immune functions and health maintenance. Long-term masticatory stimulation increases the rate of unstimulated salivary secretion and volume [4]. Therefore, long-term masticatory stimulation may affect various salivary chemical components.

The metabolome is located at the most downstream point of the central dogma; thus, it directly reflects the ever-changing biological phenomena because of its close proximity to biological phenotypes compared to genes and proteins [5]. Therefore, it is the most effective target of analysis for elucidating the causal relationship between various genetic and environmental factors and phenotypes such as disease onset. Metabolomics has been used in the medical and biochemical fields to understand the holistic view of metabolites perturbed by various diseases and treatments. The link between different health statuses and salivary metabolomic profiles has been investigated, including cancer, cognitive dysfunction, and physiological parameters [6,7,8,9,10,11]. Furthermore, unstimulated and stimulated saliva show different metabolomic profiles [12]. This result indicates that mastication may affect salivary metabolomics. However, the effects of long-term chewing on unstimulated salivary metabolomic profiles have not yet been evaluated.

The benefits of long-term chewing of sugar-free gum are attributed to increased mastication and salivation. In addition, chewing sugar-free gum yields oral health benefits, including clearance of food debris, reduction in oral dryness, increase in biofilm pH, and remineralization of enamel [13,14,15].

This study aimed to investigate the effect of long-term mastication on metabolism by comprehensively analyzing changes in unstimulated salivary metabolomic profiles using metabolomics methods in order to understand the effects of long-term mastication with a gum base under a daily and normal diet and oral hygiene.

## 2. Results

### 2.1. Overview of Quantified Metabolites

Liquid chromatography–time-of-flight-mass spectrometry (LC-TOFMS)-based salivary metabolomics successfully identified and quantified 77 cationic metabolites and 18 anionic metabolites. Furthermore, 75 frequently detected metabolites were used for subsequent analyses. Among them, 41 metabolites showed significant changes (Wilcoxon paired false discovery rate [FDR]-corrected *p*-value < 0.05) between the PRE (the first day of chewing) and POST (4 weeks after the first day of chewing) samples (Appendix A). Half of the metabolites showed Gaussian distributions; therefore, the FDR-corrected *p*-value using Student’s *t*-test is also listed (Appendix A). Volcano plots depicted the relationship between the fold change FC in PRE/POST and FDR-corrected *p*-values (Figure 1A), which revealed that the concentration of most metabolites decreased after long-term mastication. In contrast, only three metabolites (hexylamine, 2-hydroxyglutarate, and citrate) showed distinct increases (FC > 2.0). The 14 metabolites, such as o-acetylcarnitine and citrulline, showed a marked decrease (FC < 0.5) (Figure 1A). Multivariate analysis using PLS-DA yielded *R*^2^ = 0.996 and *R*^2^ = 0.910, respectively, indicating sufficient generalization ability to discriminate between PRE and POST samples (Figure 1B). In addition, our analysis identified ten metabolites with high variable importance in projection (VIP) scores (>1.5) (Figure 1C). Pathway-level changes caused by long-term mastication were also visualized (Figure 2). The pathway analysis was evaluated by the change in metabolite concentrations in each pathway (−log_10_(P)) and the centrality of the significantly changed metabolite (pathway impact). This analysis identified five pathways with significant changes and a high pathway impact (>0.5): (1) alanine, aspartate, and glutamate metabolism; (2) arginine biosynthesis; (3) D-glutamine and D-glutamate metabolism; (4) glyoxylate and dicarboxylate metabolism; and (5) arginine and proline metabolism (Figure 3).

### 2.2. Effect of Intervention on Individual Pathways

The metabolites in this map, including glutamine, glutamate, GABA, ornithine, citrulline, and arginine, showed a significant decrease after intervention (Figure 3). Glutamine and ornithine are linked to the tricarboxylic acid (TCA) cycle via glutamate, GABA, and succinic semialdehyde. The metabolites in the TCA cycle showed notable changes, and the concentrations of succinate and citrate significantly increased after the intervention. In contrast, the end products of glycolysis, such as lactate and pyruvate, decreased after intervention (Figure 3). The representative chromatograms for ten metabolites are shown in Appendix A. These results indicated the acceptable peak quality. With regard to creatine metabolism, creatine and creatinine concentrations were markedly reduced after intervention (Figure 4).

## 3. Discussion

Pathway analysis identified the following pathways showing significant differences and high-pathway-impact values: (1) alanine, aspartate, and glutamate metabolites; (2) arginine biosynthesis; and (3) D-glutamine and D-glutamate metabolism. These pathways include glutamine and arginine pathways. PLS-DA also identified glutamine as having a high VIP score (Figure 2). These pathways are related to the ornithine and glutamate cycles. 

Postprandial gum chewing has been reported to enhance diet-induced thermogenesis (energy expenditure) [16]. Chewing enhances energy production associated with metabolism caused by digestion and absorption of nutrients after meals [16]. Our study result suggests the following. The long-term mastication promoted changes in the ornithine cycle (Figure 3). We hypothesize that this would immediately metabolize ammonia caused by the production of diet-induced thermogenesis. Long-term mastication also induced metabolic flow from ornithine to glutamate (Figure 3) [17,18]. Therefore, we hypothesize that a new metabolic pathway to the TCA cycle was created via the glutamate cycle to increase the energy production of diet-induced thermogenesis in the TCA cycle. In addition, long-term mastication promoted pyruvate and lactate metabolism (Figure 3). Thus, mastication would change the glycolytic pathway and gluconeogenesis to increase ATP production during diet-induced thermogenesis.

Creatine phosphate is produced by creatine kinase and is a major source of ATP in muscles [19,20]. In our study, salivary creatine and creatinine concentrations decreased after long-term mastication, which immediately altered that metabolism compared to usual chewing (Figure 4). However, we could not measure creatine phosphate concentration that is not present in saliva but in muscle. Creatinine is the metabolite waste produced after ATP is produced by creatine phosphate or creatine and dehydration, non-enzymatically in muscles. Most of these are filtered through the renal glomerulus and excreted primarily in the urine. Our data suggested that long-term mastication immediately alters creatinine metabolism.

As described in Volcano plots, most of the metabolite concentrations decreased in the POST samples collected. The unstimulated salivary flow rate increased for the POST samples (Appendix A). This change in the unstimulated salivary flow rate is considered as one of the factors contributing to this overall decreasing trend for the metabolite. However, interestingly, metabolites (succinate and citrate) in the TCA cycle increased in the POST samples. The gum base was composed of polyvinyl acetate resin and polyisobutylene, etc. and did not contain added succinate and citrate. Therefore, we consider that the main changes in the salivary component level would be attributed to long-term mastication.

This study has several limitations. Our study only included young women with good oral-hygiene habits. Age dependence and sex differences should be investigated, and samples should be collected from subjects under various oral cavity conditions to generalize our observations. The microorganism in the oral cavity associated with systematic health status also affected the salivary properties [21,22]. Therefore, subjects in this study were in good general health, did not have periodontal disease or dental caries, and continued their normal oral hygiene routine. Subjects received professional oral prophylaxis (POP) on the day before saliva collection to reduce the effects of oral microorganisms on the salivary metabolome. Various factors, such as exercise and diet, also change the metabolomics profiles in saliva [11,23,24]. Our subjects were prohibited from exercising heavily and consuming tough foods and chewing gum except for the gum base used for this study. We could identify the effects of long-term mastication under a daily and normal diet and oral hygiene on changes in salivary metabolomic profiles. In the future, it will be necessary to search and quantitatively evaluate biomarkers related to energy using proteomics and comprehensively evaluate them together with metabolomics [25].

In conclusion, these results suggest that long-term mastication affects salivary metabolomic profiles, especially the metabolites in pathways related to energy production.

## 4. Materials and Methods

### 4.1. Subject Characteristics

Seventeen female subjects in good general health (mean age: 20.8 ± 0.3 years) were recruited from the Taiyo School of Dental Hygiene. They did not have periodontal disease or dental caries and continued their normal oral hygiene routine. In addition, subjects did not receive any medication during the study period. The Ethics Committee of the Taiyo School of Dental Hygiene reviewed and approved this research proposal, including the protocol, clinical study, and saliva collection (number: 2017-2). Subjects read and signed a written informed consent form before enrollment in the study.

### 4.2. Saliva Collection

Subjects were prohibited from exercising heavily and eating tough foods during the control and intervention periods and received professional oral prophylaxis (POP) on the day before saliva collection. A saliva sample (1.0 mL) was collected unstimulated using a Saliva Collection Aid (SCA, Salmetrics, Carlsbad, CA, USA) within 3 h of subjects cleaning their mouth after eating lunch. The SCA was placed into a collection vial chilled with crushed ice in water. Drooling saliva samples were collected through the SCA into a vial.

### 4.3. Study Design

This study was registered in the UMIN Clinical Trials Registry (ID: UMIN000031392). The experimental protocol is illustrated in Figure 5. Subjects were required to participate in a control period for four weeks before the beginning of chewing the gum base. During the control period, chewing gum was prohibited. During the intervention period, participants were instructed to chew a piece of gum base seven times a day for 10 min each time [4]. The gum base was composed of polyvinyl acetate resin and polyisobutylene, etc. and did not contain added sugar, flavor, or stabilizer. They continued to perform daily oral healthcare routines during the control and intervention periods.

### 4.4. Metabolomic Analysis

Saliva samples were pretreated as described previously [26]. In the positive-mode protocol, 10 μL of the saliva sample was mixed with 30 μL of methanol containing 149.6 mM ammonium hydroxide (1% (*v/v*) ammonia solution) and 2.5 μM of each standard compound (d_8_-spermine, d_8_-spermidine, d_6_-*N*^1^-acetylspermidine, d_3_-N^1^-acetylspermine, d_6_-*N*^1^, *N*^8^-diacetylspermidine, d6-*N*^1^, *N*^12^-diacetylspermine, hypoxanthine-^13^C_2_, _15_N, and 1, 6-diaminohexane). Following centrifugation at 20,380× *g* for 10 min at 4 °C, 30 μL of the supernatant was transferred to a fresh tube. The sample was mixed with 50 μL of water and subsequently vortexed and centrifuged at 20,380× *g* for 10 min at 4 °C. For the negative-mode protocol, each saliva sample was mixed with 30 μL of methanol containing 7 μM of internal standard (camphor-10-sulfonic acid) and subsequently vortexed and centrifuged at 20,380× *g* for 10 min at 4 °C. Subsequent manipulations were performed as described in the positive-mode protocol. Each sample (1 μL) was injected into the LC-TOFMS. The parameters of the measurement instrument were described by Shimizu et al. [27]. Briefly, metabolomic analysis of the pretreatment saliva was performed using an Agilent LC-TOFMS (Agilent Technologies, Tokyo, Japan). Raw data were analyzed using the MassHunter Workstation Software Quantitative Analysis (ver. B.08.00; Agilent Technologies). The metabolome analysis of the processed data was conducted [28]. The sequence of samples was randomized to eliminate unexpected bias, and the standard mixture and quality control (QC) samples were also measured [29]. QC samples were measured within the 12 samples. The total ion chromatography of each sample was compared to the QC samples, where the similarity of the background was ±10%. QC samples were mixed with saliva collected from healthy subjects and pretreated, as described for the saliva sample. We confirmed that the internal standard was included and that the errors in the intensity and *m/z* (±10 ppm) of the internal standard (±0.1 min) were acceptable. The relative area (e.g., the peak area of each metabolite divided by the area of the internal standard) and the small relative deviation of fluctuation (<20%) of the detected peaks in QC samples were confirmed. We prepared a standard mixture containing 106 and 59 metabolites for the positive and negative modes, respectively, and established the linearity range of each metabolite before analyzing the samples (Appendix A).

The peaks on the extracted chromatographies of the standard mixture were analyzed first, and the peaks corresponding to the standard mixture in the samples were analyzed. The absolute concentrations of salivary metabolites were calculated based on the ratio of the relative peak area in each sample to the standard mixture. The peaks whose peak size was less than the lower linearity range were treated as “not detected.” The frequently detected peaks (90% of the samples) were used for statistical analyses. Peaks without sufficient separation were eliminated from subsequent analyses.

### 4.5. Data Analysis

The Wilcoxon paired tests and paired Student’s *t*-test were used to analyze salivary metabolites between PRE (the first day of chewing) and POST (4 weeks after the first day of chewing). The Shapiro–Wilk test was also conducted to evaluate the distribution of the data. *p*-values were corrected by the false discovery rate (FDR) using the Benjamini–Hochberg method, considering multiple independent tests. The relationship between FDR-corrected P-values and the fold change (FC) of the averaged concentration of POST/PRE is depicted in a volcano plot. For multivariate analyses, partial least squares-discriminant analysis (PLS-DA) was conducted. Pathway analysis was conducted to evaluate the pathway-level changes between PRE and POST. Pathway analyses evaluated −log_10_(*Y*-axis) of the difference and pathway impact (*X*-axis) using two criteria for each pathway. The first indicated the change in metabolite concentrations for each pathway, and the second showed the centrality of the metabolite that changed significantly.

These analyses were conducted using R (version 4.0.3, R Foundation for Statistical Computing, Vienna, Austria), GraphPad Prism (version 9.2.0, GraphPad Software Inc., San Diego, CA, USA), and MetaboAnalyst (version 5.0, https://www.metaboanalyst.ca, accessed on 10 July 2020) [30].

## Figures and Tables

**Figure 1 metabolites-12-00660-f001:**
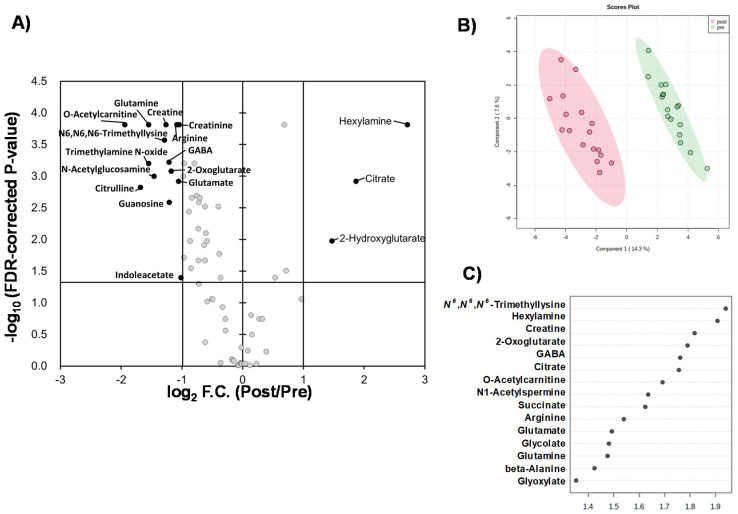
Comparisons of quantified metabolites. (**A**) Volcano plot. The *X*-axis indicates log_2_-fold change (FC) of metabolite concentrations in PRE sample divided by those of POST sample. Vertical bars indicate FC = 0.5 and 2.0. The *Y*-axis indicates −log_10_ (FDR-corrected *p*-value). The horizontal bar indicates FDR-corrected *p* = 0.05. (**B**) Score plots of PLS-DA. X and Y axes indicate the 1st and 2nd components. The *R*^2^ and *Q*^2^ values are 0.968 and 0.856 (average of 5 times of 7-fold cross-validation), respectively, with two components. Quantile normalization of each sample was conducted, and subsequently, metabolite concentration was normalized by autoscaling to eliminate sample-dependent bias. One plot indicates one sample of PRE (red) and POST (green). The light circles indicate 95% confidential intervals. (**C**) Variable importance in projection (VIP) showing top 15 metabolites. One plot indicates a metabolite. The right box indicates the higher or lower average concentrations of 0 W and 4 W.

**Figure 2 metabolites-12-00660-f002:**
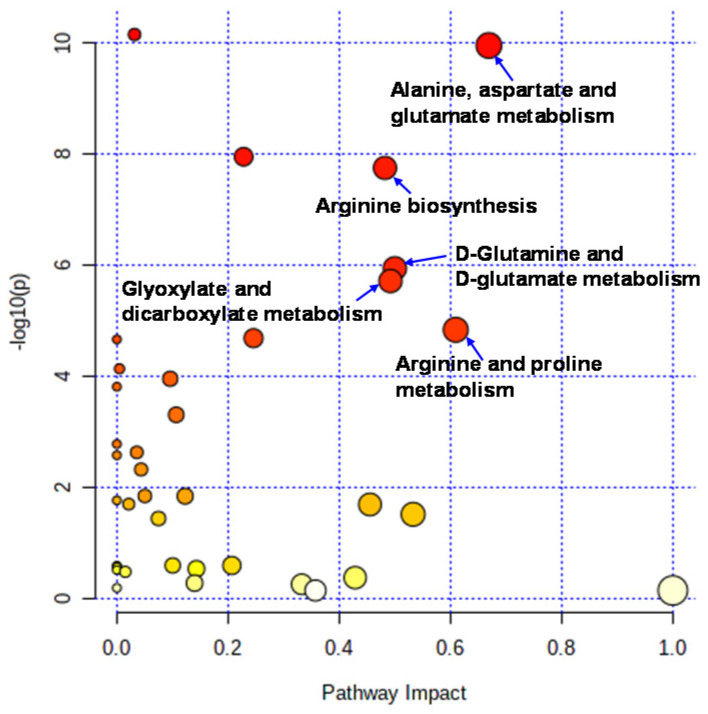
Pathways analysis. Pathway analysis was conducted to evaluate the pathway-level change between PRE and POST. The *X*-axis indicates pathway impacts. The *Y*-axis indicated −log_10_(P) of each pathway.

**Figure 3 metabolites-12-00660-f003:**
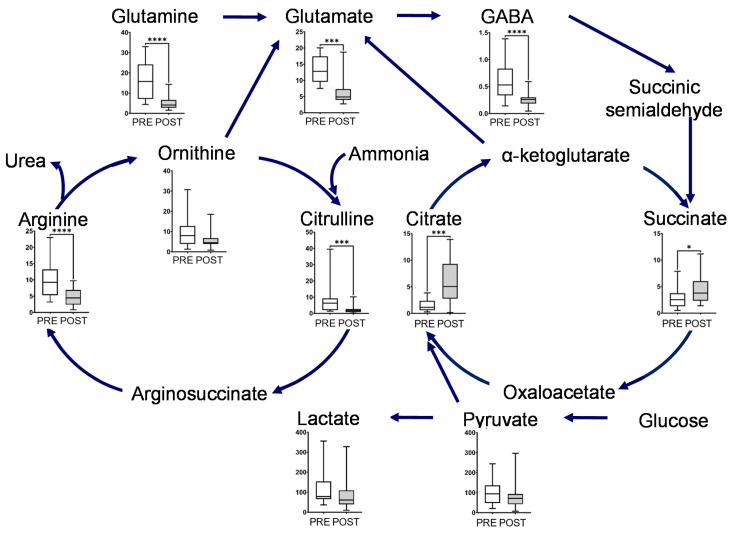
Box plot of metabolite concentration of individual pathways. Urea and glutamate cycle, TCA cycles, and glycolytic pathway. The horizontal bars of the box plots indicate the maximum, 75%, 50%, 25%, and minimum values of the data. The *Y*-axis indicates concentration (μmol/L). FDR-corrected **** *p* < 0.0001, *** *p* < 0.001, and * *p* < 0.05 by the paired Wilcoxon test.

**Figure 4 metabolites-12-00660-f004:**
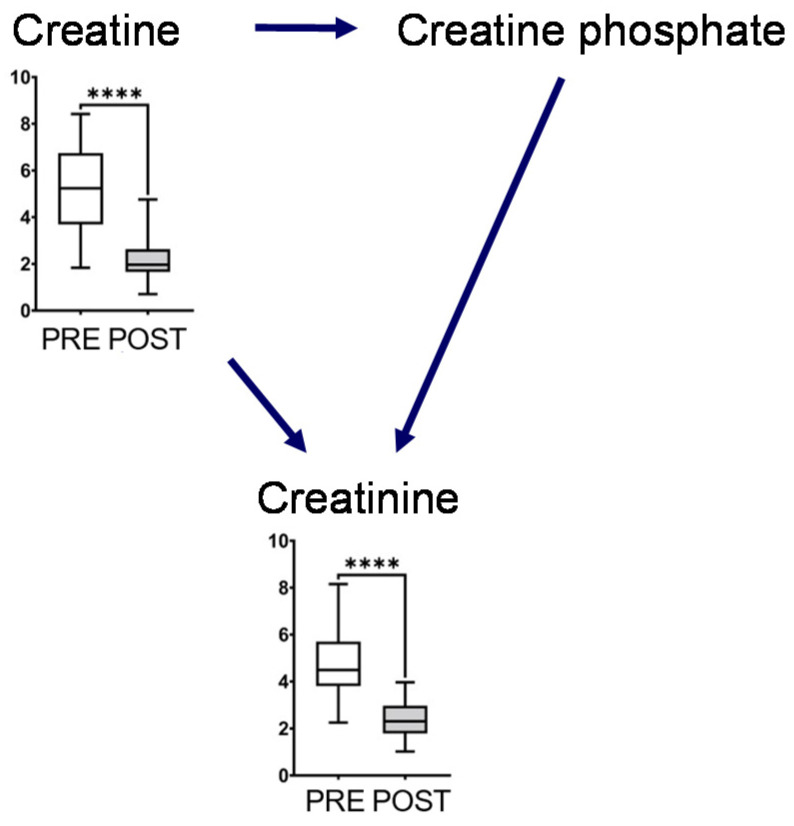
Creatine metabolism. The horizontal bars of the box plots indicate the maximum, 75%, 50%, 25%, and minimum values of the data. The *Y*-axis indicates concentration (μmol/L). FDR-corrected **** *p* < 0.0001 by the paired Wilcoxon test.

**Figure 5 metabolites-12-00660-f005:**
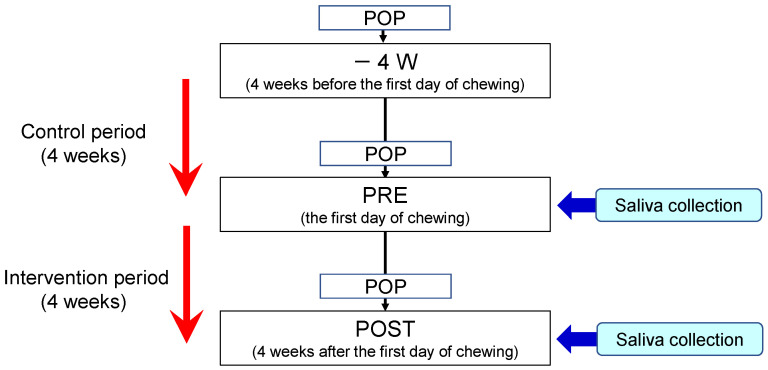
Experimental protocol. POP indicates professional oral prophylaxis.

## Data Availability

The data presented in this study are available in this article and the Appendix A.

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
