# Peer review of "Long-Term Mastication Changed Salivary Metabolomic Profiles"

_metabolites, 2022, doi:10.3390/metabo12070660_

Round 1
Reviewer 1 Report
This paper investigated effect of long-term mastication on the change of salivary metabolomics profiles and described several important metabolites related with energy production induced by the mastication. Although the research subject was rather interesting, I believe the manuscript does not fit to the protocol type of the journal “Metabolites”. The manuscript lacks detailed description in the metabolomic analysis of saliva samples and how the pathway analysis was conducted.
In terms of improving the manuscript, I suggest that authors need to provide the data of the linearity range for 106 and 59 standard mixture samples. Additionally, representative LCMS chromatogram of 10 metabolites described in Fig. 4 should be presented to clarify the peak quality of polar metabolites separated by a C18 reverse column.
Author Response
Thank you for your careful review and valuable comments and suggestions. We have attempted to address each concern and revised the manuscript accordingly. Changes to the manuscript are shown with yellow marker highlighter. According to the template of Metabolites, we have revised the order of sections in the main text as follows: (1) Introduction, (2) Results, (3) Discussion, and (4) Materials and Methods.
Our responses to your comments are given below. 
This paper investigated effect of long-term mastication on the change of salivary metabolomics profiles and described several important metabolites related with energy production induced by the mastication. Although the research subject was rather interesting, I believe the manuscript does not fit to the protocol type of the journal “Metabolites”. The manuscript lacks detailed description in the metabolomic analysis of saliva samples and how the pathway analysis was conducted.
Thank you for your valuable comments. We have described the metabolomic analysis of saliva samples and the pathway analysis methods in the revised manuscript (Lines 210-222, Lines 81-84, and Lines 256-259).
In terms of improving the manuscript, I suggest that authors need to provide the data of the linearity range for 106 and 59 standard mixture samples.
Thank you for the helpful suggestion. We have provided the required data in Supplementary Table S2 (Line 238).
Additionally, representative LCMS chromatogram of 10 metabolites described in Fig. 4 should be presented to clarify the peak quality of polar metabolites separated by a C18 reverse column.
Thank you for the excellent recommendation. We have added a representative LCMS chromatogram of the 10 metabolites described in Figure 4 in Supplementary Figure S2 and added the relevant details in the Results section (Lines 111-112).
Author Response
Thank you for your careful review and valuable comments and suggestions. We have attempted to address each concern and revised the manuscript accordingly. Changes to the manuscript are shown with yellow marker highlighter. According to the template of Metabolites, we have revised the order of sections in the main text as follows: (1) Introduction, (2) Results, (3) Discussion, and (4) Materials and Methods.
Our responses to your comments are given below.
1. Was wondering why the study is mainly focused on the metabolomics, has there been any attempt to monitor on the proteomics. Please describe the reasoning in the introduction.
Thank you for your valuable comments. We have added the reasons for focusing on metabolomics in the Introduction (Lines 41-45 and Lines 57-60) and the next attempt for Combining proteomics and metabolomics in the Discussion (Lines 173-175) sections.
2. Line 69: what does the h mean? Please replace it with hour.
Thank you for pointing this out. We have corrected h to hours in the Materials and Methods section (Line 192).
Round 2
Reviewer 1 Report
I have reviewed the revised manuscript of which contents are enhanced in terms of data reliability for the result and interpretation of the subject. I believe that the revised manuscript has an appropriate quality for the publication of the corresponding journal. One thing I would like suggest is stated below.
For the future research based on metabolomics, I suggest that authors need to carefully produce metabolite features and quantitative results according to the polarity of interests which are critical in contributing to the scientific research society.